# Direct Interaction of miRNA and circRNA with the Oncosuppressor p53: An Intriguing Perspective in Cancer Research

**DOI:** 10.3390/cancers13236108

**Published:** 2021-12-03

**Authors:** Anna Rita Bizzarri, Salvatore Cannistraro

**Affiliations:** Biophysics and Nanoscience Centre, Università della Tuscia, 01100 Viterbo, Italy; bizzarri@unitus.it

**Keywords:** microRNA, circRNA, p53

## Abstract

**Simple Summary:**

MicroRNAs and circular RNAs, which are single-stranded non-coding RNAs, play a key role as regulators at post-transcriptional level. Abnormal levels or dysregulation of miRNA or circRNA are linked to several cancerous pathologies. Starting from the evidence that some miRNAs and circRNAs are involved in the regulatory networks of the tumor suppressor protein p53, the possibility that a functional inhibition of p53 could arise from a direct interaction between p53 and oncogenic miRNAs or circRNAs was explored. Along this direction, the experimental evidence of the interaction between p53 and miRNAs and/or circRNAs is reviewed and discussed in connection with the development of new anticancer strategies.

**Abstract:**

MicroRNAs (miRNAs) are linear single-stranded non-coding RNAs oligonucleotides, widely distributed in cells, playing a key role as regulators of gene expression at post-transcriptional level. Circular RNAs (circRNAs) are single-stranded RNA oligonucleotides forming a covalently closed continuous loop, which confers them a high structural stability and which may code for proteins or act as gene regulators. Abnormal levels or dysregulation of miRNA or circRNA are linked to several cancerous pathologies, so that they are receiving a large attention as diagnostic and prognostic tools. Some miRNAs and circRNAs are strongly involved in the regulatory networks of the transcription factor p53, which plays a pivotal role as tumor suppressor. Overexpression of miRNAs and/or circRNAs, as registered in a number of cancers, is associated to a concomitant inhibition of the p53 onco-suppressive function. Among other mechanisms, it was recently suggested that a functional inhibition of p53 could arise from a direct interaction between p53 and oncogenic miRNAs or circRNAs; a mechanism that might be reminiscent of the p53 inhibition by some E3 ubiquitin ligase such as MDM2 and COP1. Such evidence might deserve important implications for restoring the p53 anticancer functionality, and pave the way to intriguing perspectives for novel therapeutic strategies. In the present paper, the experimental evidence of the interaction between p53 and miRNAs and/or circRNAs is reviewed and discussed in connection with the development of new anticancer approaches.

## 1. Introduction

MicroRNAs (miRNAs) and circular RNAs (circRNAs) are abundant, endogenous oligonucleotides belonging to the non-coding RNA families [1,2,3,4]. Although these nucleotides were initially thought to be junk DNA, it was then demonstrated that they are largely involved in a number of cellular processes in humans and, more generally, in animals, plants, and viruses [5,6].

miRNAs constitute a class of evolutionally conserved, linear single-stranded, small oligonucleotides (approximately 20–24 units). miRNAs target messenger RNA (mRNA), playing a key role as regulators of gene expression at post-transcriptional level and participating in many physiological processes, such as differentiation, cell growth, apoptosis, etc. [2,6]. Furthermore, it has emerged that miRNAs can operate through other mechanisms, many of them still to be fully clarified [7]. Dysregulation or abnormal expression of miRNAs may result in the development of a disease or in a malignant transformation [8]. Indeed, in cancer cells, miRNAs are often heavily dysregulated and overexpressed, and they may also act as oncogenes [9]. For these reasons, miRNAs are receiving a large amount of attention as diagnostic and prognostic markers [10,11].

circRNAs are single-stranded RNA oligonucleotides forming a covalently closed continuous loop between their 3′ and 5′ ends, displaying a high structural stability [3]. Despite growing research interest in circRNAs, a complete picture of the functional roles of circRNAs and their implications in biological processes is far from being reached. CircRNAs were identified to play important regulatory roles in cell proliferation, motility, and metastasis, as well as in cell cycle progression, angiogenesis, and apoptosis [12]. Moreover, circRNAs can act as miRNA sponges [13,14]. Abnormal levels or dysregulation of circRNAs are linked to several cancerous pathologies. For these features, circRNAs are also receiving a large amount of attention for being used as biomarkers in diagnostics [15,16].

Inside the plethora of different functions, some miRNAs and circRNAs were found to be strongly involved in the regulation of the p53 protein network. p53 is coded by the TP53 gene, which is part of a highly conserved family of genes and is a powerful transcription factor with a central role in the cell functionality, becoming essential in preventing inappropriate cell proliferation and in maintaining genome integrity [17]. Indeed, p53 is often mentioned as a tumor suppressor protein. p53 has a short half-life, and its level in normal conditions is kept low through its negative regulators, such as mouse double minute 2 homolog (MDM2) and constitutive photomorphogenesis protein 1 (COP1), which selectively and dominantly target p53, leading to its proteasome degradation [18]. In response to cellular stress, such as DNA damage, hypoxia, oncogene overexpression, or viral infection, the level of p53 rises, inactivating its degradation [19]. This leads to the activation of many different promoter elements, able to modulate the expression of several target genes involved in a number of cellular processes, such as DNA repair, cell cycle arrest, senescence, and apoptosis. Mutations in the p53 tumor suppressor gene occur in more than 50% of all human tumors, making it the most frequent target for genetic alterations in cancer [20]. On such a basis, the regulation of the p53 level constitutes a crucial aspect of medicine [21] and a collection of small molecules that either restore its wild-type conformation and transcriptional activity or induce synthetic lethality to mutant p53 were developed [22,23].

Overexpression of miRNAs and/or circRNAs, as registered in a number of cancers, is often associated to a concomitant inhibition of the p53 onco-suppressive function [24,25,26,27]. Along this direction, a suggestive idea is that miRNAs and/or circRNAs could exert their function on p53 through a direct interaction to p53 with mechanisms that might be reminiscent of those responsible exerted by some ubiquitin ligases and leading to p53 inhibition. Such a hypothesis might deserve a large interest for restoring the p53 anticancer functionality, by opening, thus, intriguing perspectives for novel therapeutic strategies. On the other hand, it could also provide further insights to extend the knowledge of the p53 regulatory network. In the present paper, the experimental evidence of the interaction between p53 and some circRNAs and/or miRNAs are reviewed and discussed in connection with the development of new anticancer approaches, paying also attention to underlying molecular mechanisms. In this respect, the possibility to localize the region of p53 involved in the binding with miRNAs or circRNAs deserves a high interest for providing a deeper view on the involved molecular processes.

## 2. General Properties of miRNAs, circRNAs, and p53

### 2.1. Overview of miRNAs

Human genome encodes more than 2600 mature miRNAs which are widely distributed in cells [28]. The name of a specific microRNA is usually obtained by the prefix “miR” followed by a number, indicating the order assigned to the experimentally confirmed miRNA. The emerging model of miRNA biogenesis involves a few steps [4]. Genes for miRNAs are transcribed to a primary miRNA (pri-miRNA) by RNA polymerase II, and composed of a 5′ capped and 3′ polyadenylated tail. The pri-miRNA is processed within the nucleus to a precursor miRNA (pre-miRNA) by the microprocessor complex, consisting of the RNase III enzyme Drosha4, and a double-stranded RNA-binding protein Pasha/DGCR8. The resulting pre-miRNAs are approximately 70–100 nucleotides in length, folded into imperfect stem-loop structures. The pre-miRNAs are then exported from the nucleus into the cytoplasm by the Exportin 5 and Ran-GTP complex. Once in the cytoplasm, the pre-miRNAs undergo an additional processing step by the RNAse III enzyme Dicer generating the miRNA, a double-stranded RNA, with an average length of 22 nucleotides [1]. After strand separation, one of the double strands becomes a mature miRNA molecule incorporated into the RNA-induced silencing complex (RISC), while the other one is often degraded or plays a functional role in the regulation of miRNA homeostasis. Through the mediation of Argonautic proteins and other components of RISC, miRNAs bind the complementary sequence within mRNA transcripts. The RISC complex functions by perfectly or imperfectly matching with its complementary target mRNA. miRNA nucleotides 2–8 (‘seed’ region) are thought to be essential for base pairing with mRNA. This RISC complex induces target mRNA degradation or translational inhibition or sequestration of mRNA from translational machinery, leading to a decreased expression level of target proteins with profound consequences on cellular life. Besides the complementary base-pair interaction, a class of non-canonical miRNA interactions is emerging, including imperfect seed matches or protein molecules [29].

### 2.2. Overview of circRNAs

circRNA species reported in human tissues have a size ranging from very small (100 nt) to more than 4 kb [30]. Generally, owing to their dimension, the structural characterization has some difficulties. circRNAs are identified by a number or by a name usually related to their origin. circRNAs are evolutionarily conserved and they have a tissue-specific expression. circRNAs deriving from exons are the most abundant and are generated from precursor mRNA mainly via a back-splicing process, in which the 3′ end of exon ligates to the 5′ end of its own forming a closed structure [31]. Alternatively, circRNAs can also arise from intronic and intergenic genomic regions [32]. Since the lack of the tails conferring a high resistance to degradation by exonuclease RNase R, circRNAs are characterized by a longer half-life with respect to linear RNAs. For this reason, they can accumulate in many cell types; this accumulation could be relevant in aging, especially for the brain [33]. circRNAs are principally located in the cytoplasm; however, they can also be found in nuclei and in the extracellular space [34,35]. circRNAs are implicated in multiple molecular processes, by exerting their function through different mechanisms. CircRNAs bind and interact with transcription factors to regulate the transcription process, and they can function as an miRNA sponge with a consequent regulation of the related processes [36], with sponging activity being largely involved in cancers [37]. Among other functions, there is emerging evidence that some circRNAs can bind proteins, serving as protein scaffolds, affecting protein decay or accumulation [38,39].

### 2.3. Overview of p53

p53 is a homotetramer, with each unit composed of 393 amino acids, including an intrinsically disordered N-terminal transactivation domain (TAD), a structured DNA-binding domain (DBD), and an intrinsically disordered C-terminal regulatory domain (CTD) containing a tetramerization domain connected via a flexible linker [40]. The structure of the DBD portion of p53, as obtained by X-Ray, conjugated with consensus DNA (derived from 1TUP entry from Protein Data Bank (PDB) pdb [41]), is shown in Figure 1A. DBD is composed of two antiparallel β-sheets of four and five strands, respectively, forming a β-sandwich structure. The Zn ion is tetrahedrally coordinated to the side-chains of Cys176, His179, Cys238 and Cys242, forming a Zn-finger motif, which is connected to the L2 and L3 loops [42]. The binding of DBD to DNA occurs within L1 and L3 loops in a region, which is conventionally chosen to be the northern part of the molecule.

The modular structure of p53, combined with the presence of large intrinsic disordered regions, provides a high conformational adaptability, which facilitates its interaction with a myriad of partners; this also allows a modulation of binding affinities [43]. However, the partially disordered character of p53 prevents its full structural characterization by X-Ray. The structure of full-length p53 monomer was here modeled by following the procedure reported in ref. [44]. In particular, we used the I-TASSER Suite, which is a software developed for high-resolution protein structure prediction and refinement [45]. Briefly, I-TASSER Suite was applied using the sequence and the B chain of the 1TUP entry of PDB [41]. Model 1 characterized by a medium confidence with a C-score −2.37 and a Template Modeling (TM) score of 0.44–0.14, selected as the best predicted structure, is shown in Figure 1B, together with the main domain structure.

Updated evidence indicates that both the DBD and CTD possess the capability of binding nucleic acids. More specifically, DNA preferentially binds to the DBD portion, in agreement with physiological functionality, while RNA more probably binds to CTD. As already mentioned, in non-stressed conditions, p53 levels are kept low [46]. Under a wide range of stress, p53 is activated, coordinating a complex cellular response finalized to provide a response to the stress [47]. Although the real mechanisms are extremely complex and a complete view of p53 activation is still missing [48], it can be divided into three main sequential steps: (i) p53 stabilization, (ii) binding to DNA (named p53 responsive element) and (iii) transcriptional activity. Phosphorylation of p53 represents the first step of its stabilization, following different stress signals, and it can act by a broad range of kinases. Acetylation mainly has a positive impact on the transcriptional activity of p53, while methylation can both positively and negatively regulate p53. The regulation of p53 widely occurs through post-translational modifications, including ubiquitination, phosphorylation, acetylation, methylation, at several different sites, mainly located at the TAD and CTD portions [46].

The most critical negative regulator of p53 is the E3 ubiquitin ligase mouse double minute 2 (MDM2), which directly binds to p53, mediating its ubiquitination-dependent degradation [49,50]. At the same time, MDM2 itself is a p53-regulated gene, forming an autoregulatory negative feedback loop with p53 [51]. In addition, MDM4 (also called MDMX) identified as a protein sharing structural homology with MDM2, can work alone or together to MDM2, to inhibit and degrade p53 [52]. The main binding site of p53 for MDM2 and MDM4 is the TAD region of p53 [53]. Both MDM2 and MDM4 have central acidic domains that engage in second-site interactions with the DBD by inhibiting DNA binding, while a secondary binding between MDM4 and DBD was evidenced [54,55,56]. Another important protein involved in the downregulation of p53 is COP1, a RING finger protein which targets p53 and drives its ubiquitination, with subsequent proteasome degradation [57]. Inactivation of the p53 tumor suppressor is an extremely frequent event in tumorigenesis. In most cases, the mutated p53 gene gives rise to a stable mutant protein. Mutant p53 proteins not only lose their tumor suppressive activities, but they may gain new capabilities, usually termed gain-of-function (GOF), which can actively contribute to various aspects of tumor progression [58]. In case of p53 inactivation, other members of p53 family, namely p63 and p73, characterized by a similar domain organization but with a substantially extended CTD, can vicariate the oncosuppressive function of p53 by regulating cell proliferation, differentiation and apoptosis [59]. For example, both p63 and p73 can bind to the p53R175H oncogenic mutant, with consequent abrogation of the protective function of the latter [60,61].

## 3. Interaction between p53 and miRNA

### 3.1. Overview of the Interplay between p53 and miRNAs

There is evidence demonstrating that miRNAs may strongly contribute to the regulation of p53, acting along different lines of action [27,62,63]. miRNAs can negatively regulate p53 level and function through the binding to 3′UTR of p53 mRNA, such an activity being susceptible to promote tumorigenesis [24,64]. miRNAs can target p53 regulators, such as MDM2, MDM4 and COP1, thus enhancing the p53 tumor suppressor activity [62]. Conversely, p53 can regulate the expression of miRNAs, as well as their maturation process. Furthermore, complicated interactions exist between p53 and miRNAs in regulating different pathways involved in the immune response [27]. For instance, overexpression of p53 was reported to correlate with an increased expression of miR-34a, leading to a decreased expression of programmed cell death 1 ligand (PD-L1) in tumor cells [65]. Additionally, the expression level of circulating miR-221 in the plasma was reported to correlate with the protein level of p53 in colorectal cancer [66]. In the framework of GOF, some p53 mutants upregulate the expression of oncogenic miRNAs or directly binds to the promoter of tumor suppressive miRNAs, with consequences on the genesis and progression of tumors [67]. The increasing number of identified miRNAs, together with a progressively deeper understanding of the p53 network, could lead to discover novel mechanisms by which miRNAs may regulate the p53 functions and activity. Although the knowledge of the miRNA-p53 interplay is largely uncompleted, it is now well-ascertained that miRNAs represent a class of regulators of p53 and, at the same time, p53 can regulate miRNAs expression and maturation, with these mutual influences having some relevance on the p53 network functionality.

### 3.2. Direct Interaction between p53 and miRNA

Within the plethora of the assessed mechanisms regulating the interplay between miRNAs and p53 functions, it was recently proposed that miRNAs could directly bind to p53, with possible consequences to the p53 tumor suppressor activity. At the moment, there is evidence for a direct interaction with p53, with three different miRNAs: mir21-3p, miR23-5p and miR4749-5p [68,69,70]. In the following, the experimental evidence on the interaction between p53 and each of these three miRNAs will be presented and discussed in connection with their possible biological effects. A particular attention will be devoted to extract information on the structure of miRNA as well as on the p53 regions involved in the related interaction. With such an aim, computational approaches will be reviewed when available, applied if possible, or suggested for future studies.

#### 3.2.1. Interaction between p53 and miR4749-5p

miR4749-5p is formed by 21 nucleotides, whose sequence is UGCGGGGACAGGCCAGGGCAUC and it is located at chromosome 19 of the human genome. Although there is evidence of the involvement of miR4749-5p in different signaling processes, clear insights on the role and function of miR4749-5p is still missing. It was suggested that miR4749-5p targets long noncoding RNA MAPKAPK5-AS1 and is involved in the signaling of Replication factor C subunit 2, a DNA repair-related gene, with therapeutic effects in glioblastoma multiform [71,72]. Interestingly, miR4749-5p is characterized by a high sequence similarity with the DNA Response Element of the p53 family. Such a finding led us to hypothesize a binding of miR4749-5p to p53, with possible effects on the p53 functionality. Accordingly, a direct interaction between p53 and miR4749-5p was investigated using a combination of different approaches, including fluorescence, Foerster Resonant Energy Transfer (FRET) and computational modeling [70].

The interaction between p53 and miR4749-5p was first studied by Fluorescence spectroscopy, a sensitive and powerful tool able to investigate in bulk the interactions between molecules bearing intrinsic or extrinsic fluorescence probes [73]. We focused on the possible binding of miR4749-5p to the DNA binding region of p53. With such an aim, a DBD containing a lone Trp residue (Trp146, as intrinsic fluorescent probe) was used. Experiments were carried out at an excitation wavelength of 295 nm, where practically only the lone Trp146 is excited.

Fluorescence emission of DBD alone, shown in Figure 2 (black line), reveals a maximum at 346 nm, which is indicative that Trp146 is fully exposed to the solvent [73]. The addition of progressively higher concentrations of miR4749-5p to the DBD solution yields a decrease of the fluorescence intensity, without any shift of the maximum (see Figure 2, colored lines). The observation of the fluorescence quenching indicates the formation of a complex between miR4749-5p and DBD without the involvement of the Trp146 residue of DBD. To assess the specificity of the interaction, time-resolved emission fluorescence was applied. The measured average lifetime of DBD alone (2.79 ± 0.02) × 10^−9^s and of DBD in presence of miR4749-5p (1:1 ratio) × (2.82 ± 0.02) × 10^−9^s resulted to be almost the same, which clearly indicates the formation of a static complex between DBD and miR4749-5p [73]. On such a basis, the intensity of the florescence intensity was analyzed data in the framework of the Stern–Volmer equation, which predicts a linear dependence for the ratio F_0_/F (here F_0_ and F are the fluorescence emission intensity, detected at 346 nm, of DBD alone and in presence of miR4749, respectively) as a function of concentration of the quencher. The linearity of F_0_/F vs the concentration of miR4749-5p is shown in the Stern–Volmer plot shown in the inset of Figure 2. The corresponding slope provides an affinity constant, K_A_, of (1.68 ± 0.06)·10^5^M^−1^ for our DBD/miR4749-5p-specific complex [73]. Such a value is consistent with those usually detected for complexes involving p53 [74,75].

In order to extract structural details to better characterize the interaction between DBD and miR4749-5p, an FRET experiment was performed [70]. FRET is a fluorescence-based technique useful to investigate intra- and inter-molecular distances among biomolecules bearing suitable fluorescent probes, acting as donor and acceptor, respectively [76]. In the analyzed system, Trp146 of DBD constitutes the donor (D) and the Atto390 dye, bound to the 5′ end of miR4749-5p, provides the acceptor (A), with these probes representing a very appropriate D-A couple for FRET [69]. The value of the energy transfer efficiency between D and A, calculated from the measurement of the fluorescence intensity at 346 nm, according to a 295 nm excitation, led us to estimate an average distance between D and A of 3.8 ± 0.2 nm [70]. Although such a value does not allow to univocally determine where the miR4749-5p binds to DBD, it can significantly help to find out the possible binding sites. With such an aim, a computational approach was applied. First, the secondary structure of miR4749-5p was derived by submitting its sequence to software RNAFOLD, online [77]. Such an algorithm, used under default parameters, minimizes the free energy folding. The obtained best model of the secondary for miR4749-5p, shown in Figure 3A, is characterized by a rather high prediction score, as witnessed by the presence of several high pair-base probability occurrences, as labeled by red spots. The tertiary structure was predicted starting from the best model for the secondary structure, including the corresponding dot-bracket notation, and using, online, the RNACOMPOSER software [78]; the resulting model for the tertiary structure of miR4749-5p, shown in Figure 3B.

Successively, a computational docking between the tertiary structure of miR4749-5p (shown in Figure 3B) and the DBD structure derived from 1TUP PDB entry (shown in Figure 1A) was applied using HDOCK, a docking hybrid algorithm combining template-based modeling and ab initio free docking [79]. The docking score was used to estimate the protein-ligand binding free energy. The obtained complexes were then screened by selecting those ones whose D-A distance is consistent with the value experimentally detected by FRET. A further refinement in terms of free energy binding was finally performed [70]. The resulting best model for the DBD-miR4749-5p complex is shown in Figure 4. Notably, miR4749-5p binds at the top of DBD, i.e., at the p53 region involved in the DNA binding. This finds a correspondence with the sequence similarity of the miR4749-5p with the DNA Response Element of p53.

More specifically, the interaction between miR4749-5p and DBD involves the L_2_ and L_3_ loops and the Zn-finger motif of DBD. It could then be hypothesized that miR4749-5p might hamper the binding of p53 with DNA, with a somewhat impairing of the p53 oncosuppressive activity (see Table 1).

Furthermore, the binding of miR4749-5p to DBD partially involves the H_1_ helix of DBD, which is responsible for the DBD dimerization [40]. Accordingly, miR4749-5p might also impair the oligomerization of p53. Globally, these results indicate that the interaction of miR4749-5p with p53 could affect the p53 functionality by acting along two possible different mechanisms, both of them being susceptible of leading to the inhibition of the oncosuppressive function of p53.

#### 3.2.2. Interaction between p53 and miR21-3p 

MicroRNA-21 (miR21-3p) is formed by 21 nucleotides with sequence UGCGGGGACAGGCCAGGGCAUC and it is located at chromosome 17 of the human genome. miR21-3p is among the most abundant miRNAs and one of the first identified mammalian miRNAs [83]. Furthermore, its mature sequence is highly conserved through evolution. It is practically expressed in all cells, where it performs vital regulatory roles in health and disease [84]. miR21-3p was shown to be a powerful factor that inhibits apoptosis and promotes survival in ischemia [85]. Additionally, it is one of the earliest identified cancer-promoting oncomiRs, found to be overexpressed or dysregulated in a number of human cancers (such as breast, colorectal and pancreatic cancers) [13,14,15,16]. More generally, miR21-3p is implicated in both neoplastic and non-neoplastic pathologies through many of its gene targets, associated with proliferation, apoptosis angiogenesis, metastasis and invasion. It was recently evidenced that miR-21 regulates ischemic neuronal injury via the p53/Bcl-2/Bax signaling pathway, suggesting a novel mechanism of miR-21 in regulating cerebral ischemic injury [64]. Although miR21-3p has no specificity to any one disease, it is frequently used as a generic biomarker of cancers and heart diseases in bodily fluid-based studies, with diagnostic, prognostic and therapeutic value [17]. However, its real efficiency as a prognostic biomarker is debated, since not all the results are always fully consistent [86].

The simultaneous occurrence of both p53 functional impairment and miR-21-3p overexpression, observed in some cancer types [87,88,89], led us to hypothesize a direct interaction between p53 and miR-21-3p. Such an interaction was explored by using different approaches, including fluorescence, FRET and also Atomic Force Spectroscopy (AFS); the results were refined by computational modeling. By following the same procedure used for the DBD/miR4749-5p interaction, fluorescence experiments were first carried out by adding miR21-3p to a solution of DBD containing a lone Trp residue (Trp146), with an excitation wavelength of 295 nm. Fluorescence emission intensity of DBD, showing a maximum at 346 nm, undergoes a progressive decrease upon adding a progressively higher amount of miR21-3p, without any substantial peak shift (see Figure 2 in ref. [70]). These results indicate that DBD forms a complex with miR21-3p, with Trp146 not being involved in the interaction with miR21-3p. The specificity of the interaction between DBD and miR21-3p was then assessed by time-resolved emission fluorescence upon an excitation of 295 nm. The average lifetime of DBD alone (2.79 ± 0.02) × 10^−9^s and of DBD in presence miR21-3p × (2.82 ± 0.02) × 10^−9^s was found to be almost the same, supporting the formation of a specific complex between miR21-3p and DBD [69]. On such a basis, an analysis of the florescence intensity emission, detected at 346 nm, of DBD as a function of the miR21-3p concentration, was performed in the framework of the Stern–Volmer equation [69]; this allowed us to extract the affinity constant, K_A_, for the DBD/miR21 complex. We found for K_A_ a value of (1.2 ± 0.2) × 10^5^M^−1^, which reflects a moderate affinity between the partners. Furthermore, such a value is close, although slightly lower, to that found for miR4749-5p and DBD, suggesting a similar interaction between DBD and miR21-3p. The formation of a specific complex between DBD and miR21-3p represents important information which could be reinforced by extracting more details about the interaction geometry, which, in turn, may lead us to hypothesize possible effects arising from this interaction. Again, the donor is provided by lone Trp146 of DBD, while the acceptor is the Atto390 dye bound to the 5′ end of miR21-3p. Upon evaluating the energy transfer efficiency between D and A, an average D–A distance, D_DA_, of (3.9 ± 0.4) nm between Trp146 of DBD and the dye (Atto390) was determined. Such a distance could be consistent with a binding site for miR21-3p at the top of DBD [69].

To more strongly support such a hypothesis, and by taking advantage of the D-A distance, the same computational procedures developed to determinate the binding site between DBD and miR4749-5p, were here applied to the DBD and miR21-3p partners. Preliminary, the secondary structure of miR21-3p, not available, was derived by RNAFOLD [77]. The obtained model, shown in Figure 5A, is predicted with a high score, as witnessed by the presence of red spots throughout its sequence. Then, the tertiary structure was modeled by RNACOMPOSER software [78], the best models being shown in Figure 5B.

Such a model for the tertiary structure for miR21-3p was then submitted to a computational docking with DBD by the HDOCK software [79]. Upon screening the found models in terms of the D-A distance and the binding free energy, we found the best model for the complex between DBD and miR21-3p, shown in Figure 6.

miR21-3p binds to DBD at the top in a portion adjacent to the DNA binding region, with the involvement of the DBD L_3_ loop, which binds to the minor groove of DNA. Accordingly, it could be speculated that miR21-3p might partially impair the capability of p53 to bind DNA, with possible inhibition of the p53 main activity (see Table 1). We note on passing that this interaction region is close to that found for miR4749-5p, even if no substantial overlap is observed. These results might suggest that the overexpression of miR21-3p, which is observed in some tumors, could give rise to a reduction of the oncosuppressor activity of p53 due to a limitation of the p53 capability to bind to DNA. In this respect, drugs able to sequestrate miR21-3p might help to restore the p53 activity.

To further support the formation of a specific complex between DBD and miR21-3p and, at the same time, to provide more details about the interaction between DBD and miR21-3p, Atomic Force Spectroscopy (AFS) was applied. AFS is a single-molecule technique, performed by AFM equipment, operating under near-physiological conditions and suited to investigate the unbinding kinetics of complexes through the measurements of intermolecular forces [90,91]. A typical force curve detected in an AFS experiment is sketched in Figure 7.

Briefly, the tip at the end of the AFM cantilever is functionalized with the ligand (here miR21-3p, as shown in the inset of Figure 7), while the receptors (here DBD) are immobilized on the substrate. Starting from the initial positions (Point 1), the tip is approached to the surface at a given loading rate (dF/dt). When the tip contacts the substrate, the cantilever begins to deflect due to intermolecular repulsive forces, while the ligand and receptor could interact and eventually form a complex (Points 2 and 3). The cantilever is then retracted from the surface. Due to attractive interaction forces, the cantilever begins to bend downward (Point 4). When the force exerted by the cantilever overcomes the force of the complex, the complex dissociates and a sudden jump (called a jump-off) in the deflection is detected (Point 5). Finally, the tip comes back to the initial position (Point 6), from which a new force-distance cycle can start. The measurement of the jump-off allows to determine the unbinding force required to break the complex. A statistical analysis of the unbinding forces as a function of loading rate in the framework of suitable models (here we used the Bell–Evans model [92]), can provide the dissociation rate constant, k_off_, of the complex and the width of the energy barrier along the direction of the applied force x_β_. From an analysis of the force curves collected at five different loading rates, we found a k_off_ = (3.0 ± 0.7)·10^−1^ s^−1^ and x_β_ = (0.26 ± 0.09) nm; these should be discussed in connection with the results found for similar biological complexes. The x_β_ value is close to those found for some complexes involving p53 [74,93,94,95,96], suggesting that a similar interaction occurs. In other words, the interaction could follow similar molecular mechanisms. Additionally, the dissociation rate of the complex, to which corresponds a lifetime, τ of (τ = 1/k_off_) of 3.3 s, indicates that the complex is rather stable. Notably, such a value is lower than those usually detected for complexes involving RNA and proteins [97]. These results globally suggests that miR21-3p could compete with other partners of p53 and it deserves some interest even in the perspective to develop specific drugs able to strengthen the p53 capability to bind DNA.

#### 3.2.3. Interaction between p53 and mir23-5p

miR23-5p (also named miR23a) is formed by 22 nucleotides, with sequence GGGGUUCCUGGGGAUGGGAUUU and it is located at chromosome 19 of the human genome. Dysregulation of miR23-5p was reported in various human diseases, such as ischemia-reperfusion injury, coronary heart diseases, and cancers [6]. For these reasons, miR23-5p was extensively studied in different types of human cancers, where it can play various roles in the initiation, progression and maintenance of cancer cells [98]. For example, miR23-5p is often downregulated in both acute and chronic myelogenous leukemia, while in a rare form of myeloid leukemia it is upregulated. Additionally, miR23-5p is downregulated in most of the cancers occurring in the genitourinary system, such as prostate carcinoma, endometrial endometrioid adenocarcinoma and nephroblastoma [98]. Notably, a relationship was identified between the E2-signaling mechanism in regulating the activation of p53 and miR23-5p expression in liver cancer cells, suggesting a possible view to the understanding of the sex difference observed in this disease entity [99].

A recent study demonstrated that miR23-5p directly participates to the apoptosis as induced by oxidative stress [68]. Trying to unveil the underlying molecular mechanism, pull-down p53 immunoprecipitation evidenced that miR23-5p specifically and directly binds p53 in the cell nuclei of cardiomyocytes [69]. Such a binding is at the basis of the association of p53 with the promoter region of miR128, and then the enhancement of the expression of miR128, which, in turn, leads to apoptosis [69]. At the present, no other information about the interaction between p53 and miR23-5p is available. In order to further characterize this complex, a computational procedure, similar to that previously followed for miR4749-5p and miR21-3p, was applied to shed light on the interaction between p53 and miR23-5p.

Since no information about the structures of miR235p is available, preliminarily a modeling procedure for the secondary structure was applied. The secondary structure of miR23-5p, obtained by RNAFOLD [77], is shown in Figure 8A; a rather good pair-base probability prediction being reached (see red and orange spots). Successively, a model for the tertiary structure was derived by using RNACOMPOSER software [78]; the resulting best model is shown in Figure 8B.

By considering that there is no evidence about which portion of p53 interacts with miR23-5p, the possibility of an interaction with the whole p53 was taken into consideration. With such an aim, the model of p53 previously obtained, and shown in Figure 1B, was submitted to a computational docking by HDOCK [79] with the tertiary structure of miR23-5p (see Figure 9B).

An analysis of the 10 best predicted best models for the p53/miR23-5p complex revealed that in almost all the cases, miR23-5p clusters at the top of p53, in a region close to the DNA binding site. A refinement of these models, also by free energy binding analysis, allowed us to extract the best model which is shown in Figure 9A. miR23-5p (olive color) binds at the top of p53 leaving almost free the DNA binding domain. Additionally, the binding of miR23-5p does not involve the CTD portion (see the lateral view of the best complex shown in Figure 9B), which is known to have the capability to bind with other molecules and in particular with oligonucleotides (mainly RNAs). Accordingly, it could be speculated that the formation of the complex between p53 and mir23 might favor the interaction of p53 with other molecules, such as miR-128, whose role was demonstrated to be crucial to regulate the p53-mediated processes leading to cell apoptosis (see Table 1). Although further investigations are required to find out more experimental evidence, these results provide some preliminary indications, which could be useful for designing new experiments, helping to understand how the formation of the p53/miR23-5p complex could provide new insight both on the p53 regulatory network and the possible direct effect on its activity.

## 4. Interaction between p53 and circRNA

### 4.1. Overview of the Interplay between p53 and circRNA

Although a complete view of circRNA functions is still far from being reached, there is some evidence of an interplay between circRNAs and the p53 network [38,82,100]. For example, the association of circANRIL with the Pescadillo Ribosomal Biogenesis Factor 1 (PRF1) protein leads to p53 upregulation, increasing apoptosis in highly proliferating cancer cells [101]. Additionally, breast cancer progression by mutant p53 is inhibited by the circCcnb1 through the formation of the ternary complex circCcnb1-H2AX-Bclaf1 complex, inducing tumor cell death [82]. By taking into consideration the capability of circRNAs to directly interact with proteins, the binding between circRNAs and p53 represents a further regulation element of p53 whose implications may deserve a high interest from both fundamental and therapeutics points of view. In this context, it was found that some circRNAs exhibit the capability to directly bind p53, affecting the p53 signaling. In the following, the interaction between p53 and circCDR1as, circFOXO3 and circDNMT1 will be presented and discussed in connection with their implications for the cell functionality.

### 4.2. Direct Interaction between p53 and circRNAs

#### 4.2.1. Interaction between p53 and circCDR1as

circCDR1as (also named circ-0001946, or CDR1as) is generated from the gene 19 coding for Cerebellar degeneration-related protein 1 (CDR1) antisense RNA. circCDR1as is formed by 1485 nucleotides, and it is one of the most well-identified and investigated circRNAs [35,36]. A tentative model of its secondary structure, characterized by a minimum free energy of −201.20 kcal/mol, obtained by RNAFOLD [77], is shown in Figure 10. We note that the secondary structure has a high structural complexity, with some local portion having a high prediction score (red and orange colors), while the skeleton is poorly predicted (blue color).

It can act as miRNA sponge with the capability to absorb several different miRNAs [102]; with this activity exerting a wide range of effects in both physiological and pathological processes, even promoting the proliferation of cancer [103]. Along this direction, an important sponging action exerted by circCDR1as is with respect to miR-7 for which is known to have more than 70 different binding sites [36,104]. Since in many types of cancers (such as osteosarcoma, breast cancer, and colorectal can), miR-7 functions as a tumor suppressor, such a sponging activity promotes proliferation of cancer cells [103,105]. More generally, several studies demonstrated that circCDR1as is abnormally expressed in many types of tumors, such as colorectal cancer and osteosarcoma [103]. Additionally, circCDR1as is highly expressed in human brain playing a crucial role in brain development and, at the same time, it is strictly related to neurodegenerative diseases, such as tumorigenesis of glioma [106].

Very recently, Lou et al. demonstrated that p53 specifically and strongly interacts with circCDR1as, promoting an anticancer action with the inhibition of glioma tumor growth in vitro and in vivo [80]. Conversely, inactivation of circCDR1as contributes significantly to tumorigenesis of glioma. The interaction between circCDR1a and p53 was assessed by RIP-seq and RIP-qPCR assays combined with bioinformatic methods. With the aim to clarify the underlying mechanism, the interaction region between p53 and circCDR1as was investigated by testing the binding capability of different p53 constructs with circCDR1as [80]. It was found that the construct missing the DBD domain lost its capability to bind circCDR1as, indicating that DBD is directly involved in the interaction with circCDR1as. On the other hand, circCDR1as has little impact in cells where p53 is absent or mutated. Additionally, the interaction of circCDR1as with p53 prevents ubiquitination by MDM2 and subsequent degradation of p53. More specifically, the direct interaction between circCDR1as and p53 can disrupt the p53/MDM2 complex, with such an interaction likely occurring through the involvement of the DBD portion of p53, constituting the secondary binding site of p53 for MDM2. Remarkably, these results suggest a new possible strategy to target the p53/MDM2 complex, which plays a crucial role in the p53 regulation (see Table 1). Indeed, the capability of circCDR1a to stabilize p53 through decreasing ubiquitination by MDM2 represents a further way to potentiate the p53 anticancer activity.

In summary, the direct interaction between circCDR1as and p53 leads to the stabilization of p53, which can, in turn, promote an anticancer action with tumor inhibition. It is interesting to remark that the tumor suppressor role played by circCDR1as is opposite to its oncogenic function exerted, e.g., through the miRNA sponging [35]. This generally indicates a possible dual role of circRNAs depending on the context, confirming the complexity of the noncoding RNA world.

#### 4.2.2. Interaction between p53 and circFOXO3

circFOXO3 (also named circ-0006404), derived from exon 2 of Forkhead box O3 (FOXO3), is an element of the FOXO family involved in the regulation of the cell cycle, energy metabolism and tumorigenesis [107]. circFOXO3 contains 1435 nucleotides. A model of the secondary structure of circFOXO3, obtained by RNAFOLD [77] and characterized by a minimum free energy of-498 kcal/mol, is shown in Figure 11. Similarly to what found for circCRD1as, the secondary structure exhibits a high complexity. The model exhibits a rather high prediction score (red color) throughout the whole structure. However, it is slightly different from that obtained by another algorithm (see ref. [81]); such a difference is not surprising due to the extension of the system that makes predictions rather difficult.

circFOXO3, whose function partially overlaps with that of FOXO3, can increase the level of FOXO3 and, at the same time, it participates in the regulation of transcription products through a competitive endogenous RNAs network [108]. circFOXO3 has extensive and complex biological functions covering cell differentiation, apoptosis and cell cycle progression, with many mechanisms still unclear [107]. On the other hand, circFOXO3 is widely expressed in a series of malignant tumors, e.g., it is upregulated in prostate cancer and in hepatocarcinoma [109,110]. As with other circRNAs, circFOXO3 acts as a miRNA sponge with the capability to adsorb multiple miRNAs (such as miR29a-3p, miR138-5p and miR214), thus regulating the expression of downstream genes and cancer progression [107,111].

Low levels of circFOXO3 were recently found in breast carcinoma, while overexpression of circFOXO3 induces an increase of FOXO3, leading cancer cells to apoptosis [81]. With the aim to clarify the underlying molecular mechanisms, it was revealed that circFOXO3 binds to both MDM2 and p53. In particular, a computational docking found that circFOXO3 binds to the CTD portion of p53, with the involvement of the circFOXO3 region with sequence of GGGUGCCAGGCUGAAGGAUCACUG [81]. Another region of circFOXO3 likely docks with the RING-finger domain of MDM2. The binding of p53 and MDM2 to circFOXO3 facilitates the addition of ubiquitin to p53 and MDM2, with this also repressing the MDM2 ubiquitination and degradation of FOXO3. Such a process leads to an increase of the FOXO3 activity and then to a promotion of cell apoptosis [81]. On the other hand, it was hypothesized that the interaction among p53, MDM2 and circFOXO3 may introduce a further protective mechanism against mutant p53. Since MDM2 cannot bind p53 mutant, the translocation of mutant from nucleus to cytosol, an essential step for p53 degradation, is decreased with an increase of the p53 level. Therefore, the binding of circFOXO3 to both p53 and MDM2 could represent a new regulatory path of p53 able to modulate its response (see Table 1). The interplay among circFOXO3, p53 and MDM2 opens a new way to understanding the balance between p53 and MDM2, whose role is central in the tumor suppressor activity of p53. Notably, these findings further support how different regulators could interfere to regulate the mechanisms of specific partners, such as p53-MDM2.

#### 4.2.3. Interaction between p53 and circDNMT1

circDNMT1 (also named hsa_circ-102439) derives from exons 6 and 7 of DNA methyltransferase1 (DNMT1), which is an enzyme that plays a role in the regulation of tissue-specific methylation of cytosine residues [112]. circDNMT1 contains 155 nucleotides. A model of its secondary structure, obtained by RNAFOLD [77] and characterized by a minimum free energy of −20.15 kcal/mol, is shown in Figure 12.

circDNMT1 is involved in the proliferation and survival capacities of cells [82]. CircDNMT1 was found to be upregulated in all cancer cells, with a very high level in breast cancer tissues and cell lines. Expression of circDNMT1 inhibits cellular senescence and increases tumor cell proliferation and also stimulates cellular autophagy. At the same time, silencing circ-Dnmt1 inhibited cell proliferation and survival.

It was recently demonstrated that expression of circDNMT1 results in the inhibition of p53 transcription [82]. At the molecular level, it was found that circDNMT1 interacts with both p53 and AU-rich element RNA-binding protein 1 (AUF1). The interaction of circDNMT1 with p53 promotes nuclear translocation of both p53 and AUF1. The association of circDNMT1 with p53 induces cellular autophagy, while the association with AUF1 promotes nuclear localization of AUF1, which, in turn, upregulates DNMT1, leading to the inhibition of p53 expression (see Table 1). Computational approaches support that circDNMT1 has two distinct binding sites for p53 and AUF1. In particular, it was predicted that circDNMT1 could dock at the CTD region of p53 with the involvement of the sequence AACCTTCAC″AAC″AGGAAGAA. Additionally, the regions of circDNMT1 docking with the portions of AUF1 (involving the N-terminus and the glutamine-rich binding domains) were also predicted. In summary, these results indicate that the interaction of circDNMT1 with p53, eventually with the support of other biomolecules, could play a central role in cancer, and these findings might open new perspectives in the development of new p53-based strategies.

## 5. Future Perspectives and Conclusions

p53 plays a crucial regulator role in many biological processes through an extremely complex interaction network characterized by many interconnections among the various elements. Recently, it was evidenced that such a network should also include miRNAs and circRNAs, which are single-stranded non-coding RNAs oligonucleotides, widely distributed in cells, where they play several roles acting along different action lines.

The miRNA and circRNA world is characterized by a high complex behavior. In this context, the direct physical binding between p53 miRNAs or circRNAs represents a new circumscribed, but at the same time very intriguing, possibility, enlarging the overview on the p53 regulator molecular mechanisms. According to examples presently reviewed (see Figure 13), it appears that the direct interaction between p53 and miRNAs and/or circRNAs could highlights new regulatory pathways in the complex p53 network, which may shed new light on normal and pathological cell functionality. This evidence may provide a remarkable ground to both open new perspectives in the understanding of the control mechanisms underlying the p53 role and to inspire the development of new strategies for cancer therapies based on the preservation and enhancement the p53 tumor suppressor function. Up to now, the direct interaction between p53 and miRNAs or circRNAs was disclosed only for few cases; however, it could be expected that further evidence will likely come out as far as new ad hoc investigations will be carried out. On the other hand, the discovery of new miRNAs and circRNAs elements, in connection with the disclosure of their implication in many functions, including their interplay with p53, is susceptible to become a very hot research theme in the near future.

## Figures and Tables

**Figure 1 cancers-13-06108-f001:**
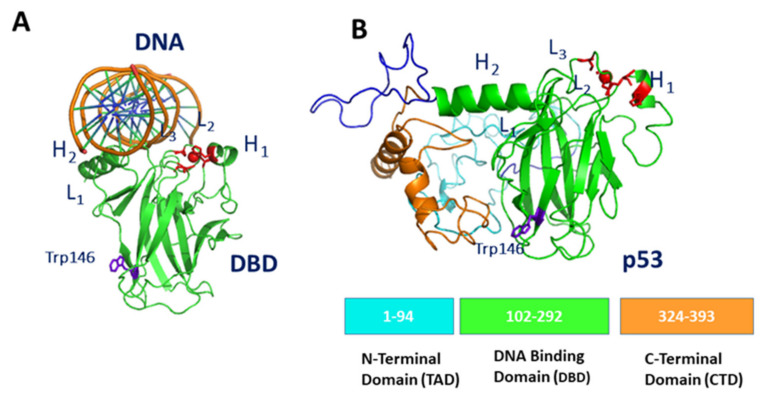
(**A**) Graphical representation of DBD complexed with DNA (as derived from chain B of 1TUP PDB entry [41]). (**B**) Graphical representation of the 3D model for the full-length monomer of p53 (see the text); at the bottom, the main domain structure is shown. In both views, Trp146 is marked in blue, while Zn and the residues forming the Zn-finger are marked in red.

**Figure 2 cancers-13-06108-f002:**
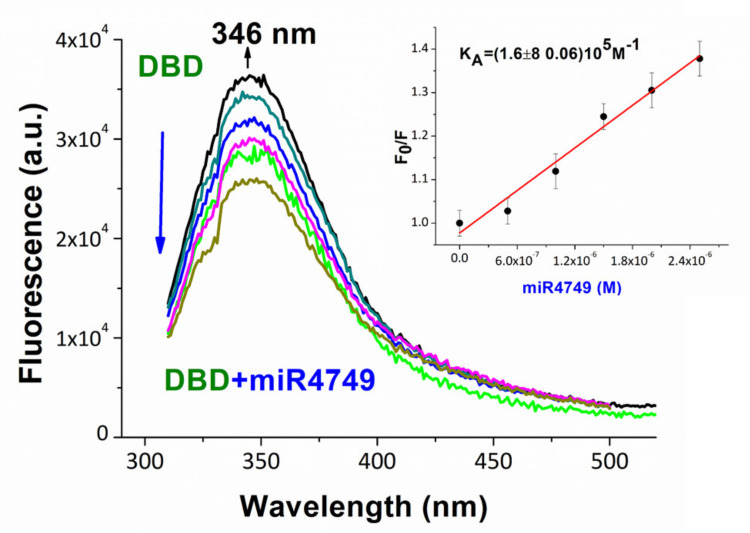
Fluorescence emission spectra of DBD at the concentration of 1 µM (black line), and upon the addition of miR4749-5p at higher concentrations ranging from 0.5 to 2.5 µM (colored lines). Samples were excited at 295 nm and the spectra were corrected for the Raman scattering of the buffer. Inset: Stern-Volmer plot of the fluorescence quenching of DBD as a function of miR4749-5p concentration, shown as black circles. The continuous red line represents the linear fit by the equation F_0_/F = 1 + K_A_*C[miRNA4749-5p]; the extracted constant, K_A_, being shown. Adapted from ref. [70].

**Figure 3 cancers-13-06108-f003:**
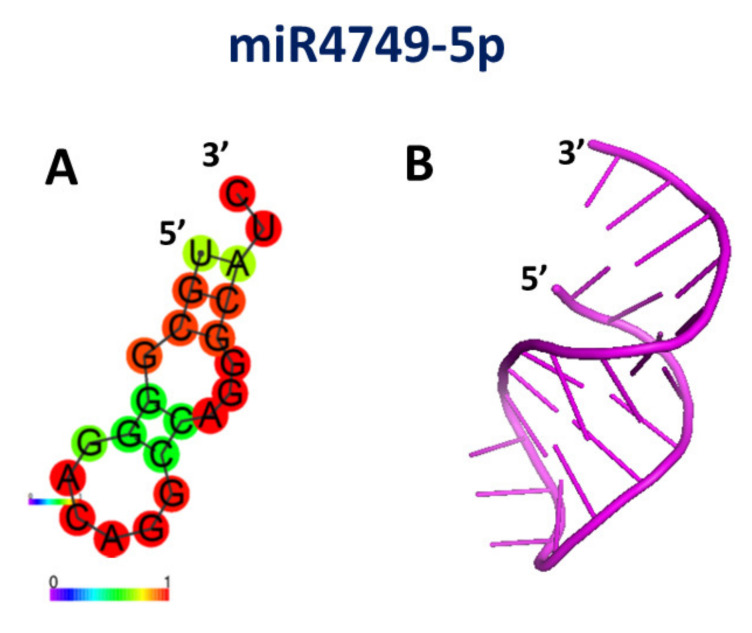
(**A**) Drawing of the best model for the secondary structure of miR4749-5p; colors of the spots indicate the corresponding pair-base probability (see the scale at the bottom). (**B**) Graphical representation of the best 3D model for the tertiary structure of miR4749-5p, as derived using the secondary structure in (**A**) together with related dot-bracket notation. Adapted from ref. [70].

**Figure 4 cancers-13-06108-f004:**
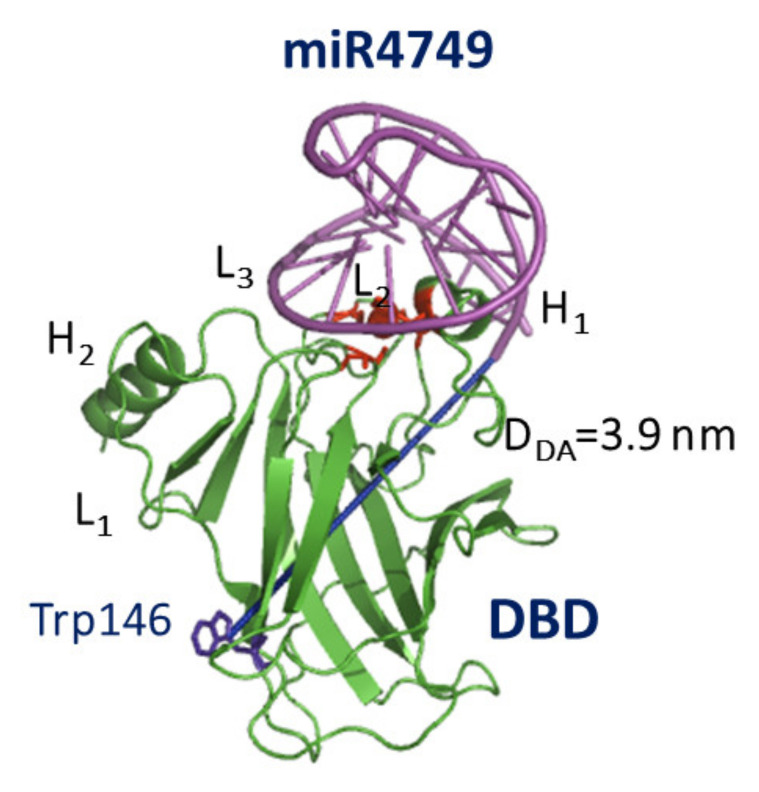
Graphical representation of the best model for the complex between DBD (green) and miR4749-5p (magenta). The initial structure of DBD was taken from the chain B of 1TUP PDB entry, while that of miR4749-5p is the model in Figure 3B). The distance D_DA_ between the center of the aromatic rings of Trp146 and the 5′ end of miR4749-5p is marked as a blue line. Adapted from ref. [70].

**Figure 5 cancers-13-06108-f005:**
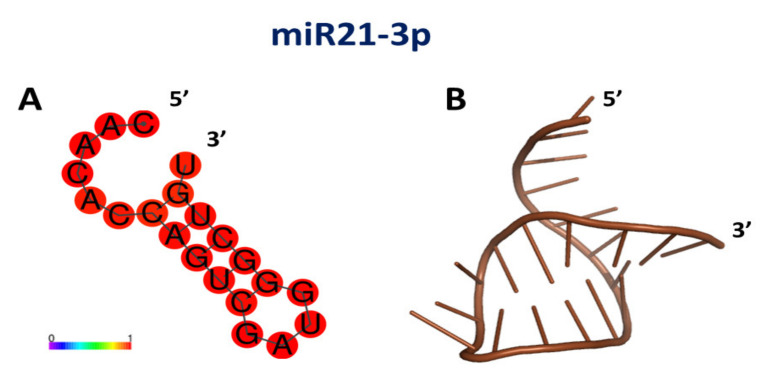
(**A**) Drawing of the best model for the secondary structure of miR21-3p; colors of the spots indicate the corresponding pair-base probability (see the scale at the bottom). (**B**) Graphical representation of the 3D best model for the tertiary structure of miR21-3p, as derived using the secondary structure in A) together with related dot-bracket notation.

**Figure 6 cancers-13-06108-f006:**
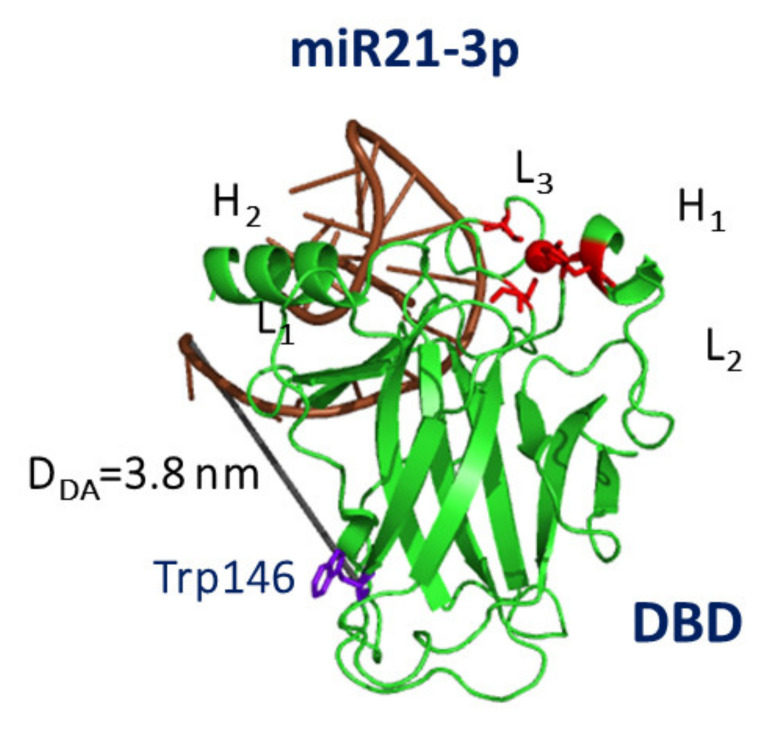
Graphical representation of the best model for the complex between DBD (green) and miR21-3p (brown). The initial structure of DBD was taken from the chain B of 1TUP PDB entry, while that of miR21-3p is the model in Figure 5B. The distance D_DA_ between the center of the aromatic rings of Trp146 and the 5′ end of miR21-3p is marked as blue line.

**Figure 7 cancers-13-06108-f007:**
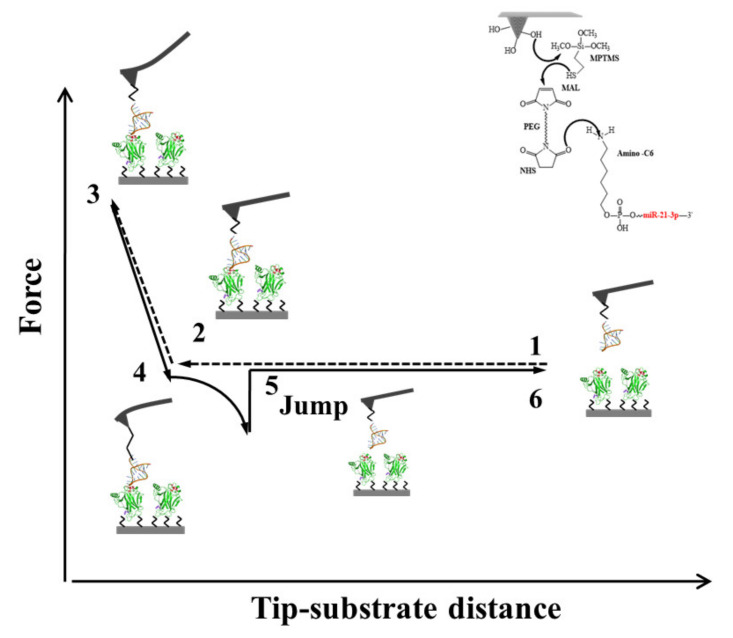
A typical approach–retraction cycle of the miR21-3p-functionalized tip over the DBD-functionalized substrate showing a specific unbinding event. Point 1: The cantilever moves toward the substrate. Point 2: The tip reaches the contact point, with the two partners eventually forming a complex. Point 3: The cantilever further moved toward the substrate, undergoing an upward deflection. Point 4: The cantilever is retracted from the substrate with a bending downward due to attractive interaction forces between the partners. Point 5: The tip undergoes a jump-off with an unbinding of the complex. Point 6: the cantilever returns to its initial position.

**Figure 8 cancers-13-06108-f008:**
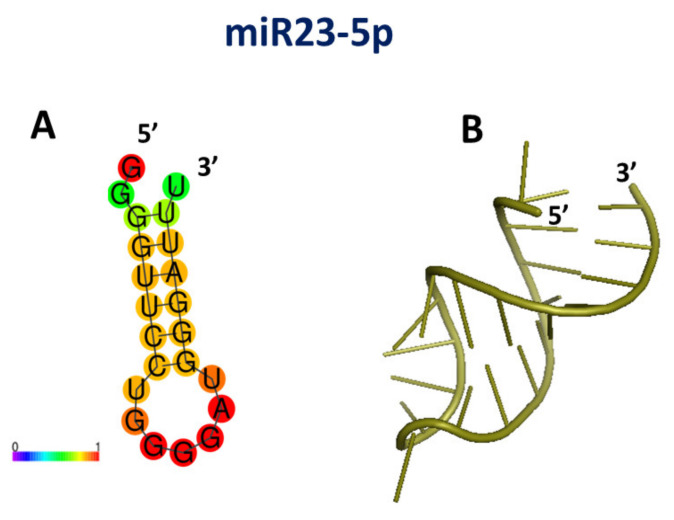
(**A**) Drawing of the best model for the secondary structure of miR23-5p; colors of the spots indicate the corresponding pair-base probability (see the scale at the bottom). (**B**) Graphical representation of the 3D best model for the tertiary structure of miR23-5p, as derived using the secondary structure in (**A**) together with related dot-bracket notation.

**Figure 9 cancers-13-06108-f009:**
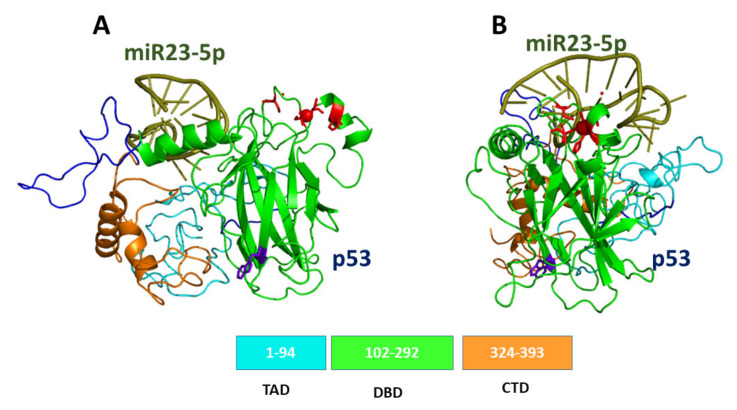
Front (**A**) and lateral (**B**) views of the graphical representation for the best model for the complex between p53 and miR23-5p (olive). The initial structure of p53 was taken from the model shown in Figure 1B, while that of miR23-5p is the model in Figure 8B.

**Figure 10 cancers-13-06108-f010:**
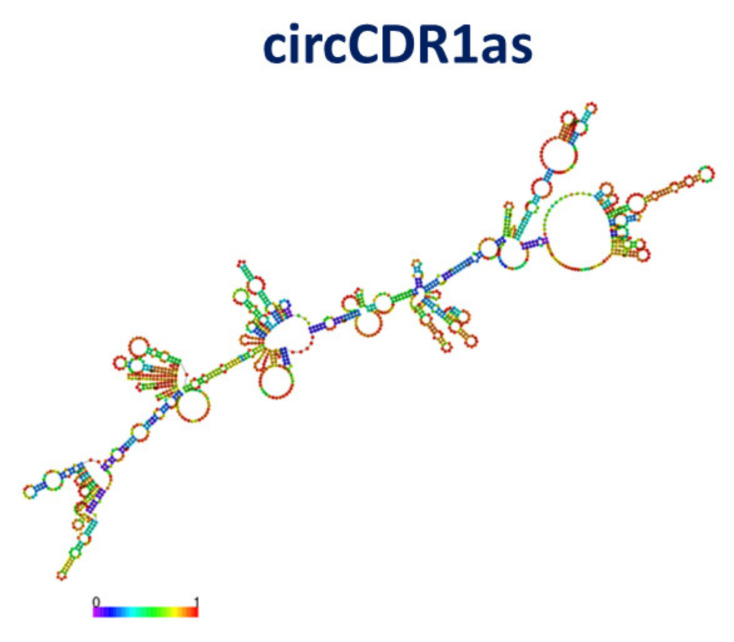
Drawing of the best model for the secondary structure of circCDR1a; colors of the spots indicate the corresponding pair-base probability (see the scale at the bottom).

**Figure 11 cancers-13-06108-f011:**
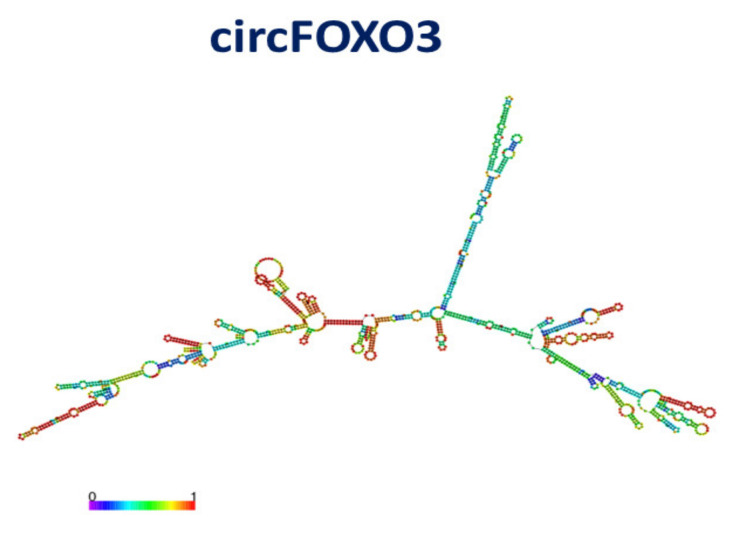
Drawing of the best model for the secondary structure of circFOXO3; colors of the spots indicate the corresponding pair-base probability (see the scale at the bottom).

**Figure 12 cancers-13-06108-f012:**
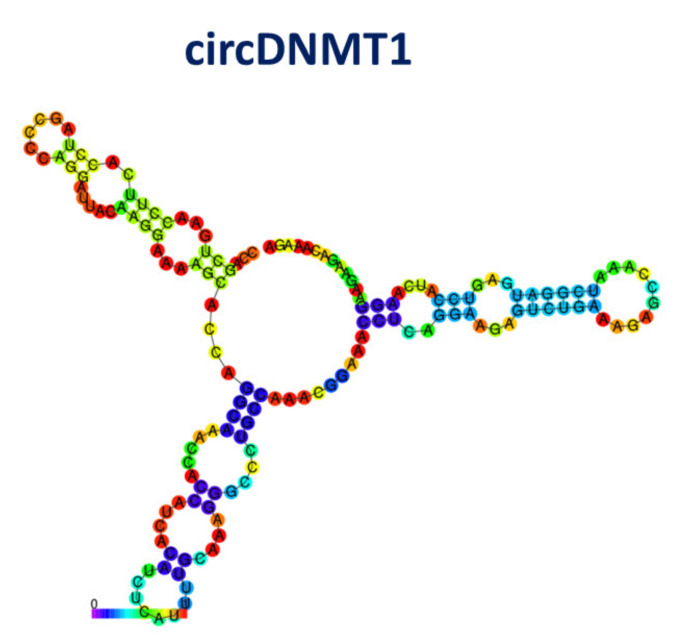
Drawing of the best model for the secondary structure of circDNMT1; colors of the spots indicate the corresponding pair-base probability (see the scale at the bottom).

**Figure 13 cancers-13-06108-f013:**
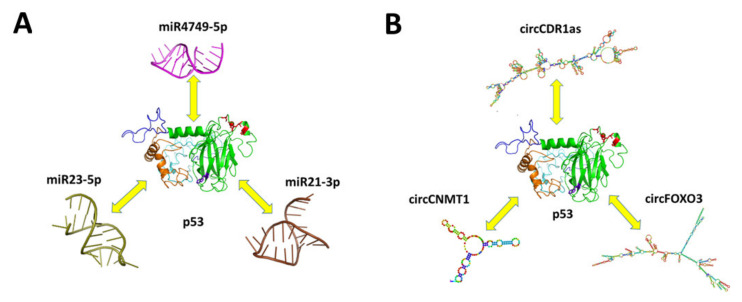
(**A**) Overview of the miRNAs directly interacting with p53. (**B**) Overview of the circRNAs directly interacting with p53.

**Table 1 cancers-13-06108-t001:** Summary of the interaction properties between p53 and the investigated non-coding RNA.

p53-RNA Interaction	Domain of p53 Involved in the Binding	Proposed Regulation Mechanism	Ref.
p53-miR4749	DBD	Inhibition of p53-DNA interaction	[70]
p53-miR21	DBD	Inhibition of p53-DNA interaction	[69]
p53-miR23	CTD	Promotion of p53-miR128 interaction, favouring apoptosis due to oxidative stress	[68]
p53-circCDR1	DBD	Disruption of the p53-MDM2 complex, favouring an anticancer action	[80]
p53-circFOXO3	CTD	Promotion of MDM2 ubiquitination, favouring cell apoptosis	[81]
p53-circCNM1	CTD	Inhibition of p53 transcription leading to tumor cell proliferation	[82]

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
