# Peer review of "Direct Interaction of miRNA and circRNA with the Oncosuppressor p53: An Intriguing Perspective in Cancer Research"

_cancers, 2021, doi:10.3390/cancers13236108_

Round 1
Reviewer 1 Report
Bizzarri et al. described experimental evidence of interaction between p53 and miRNAs and/or circRNAs. The review is interesting and clearly described. Few comments to improve this review article.
- It will be helpful to understand if there is a figure summarizing the interaction of p53 and different miRNAs as well as a figure summarizing the interaction of p53 and different circRNAs.
- Proofreading is necessary. There are many sentences with big spaces between words.
Author Response
circRNAs. The review is interesting and clearly described. Few comments to improve this review article
Thank you for your overall positive comments.
- It will be helpful to understand if there is a figure summarizing the interaction of p53 and different miRNAs as well as a figure summarizing the interaction of p53 and different circRNAs.
A new figure summarizing the interaction of p53 with both miRNAs and circRNAs, has been added to the manuscript as new Figure 13.
- Proofreading is necessary. There are many sentences with big spaces between words.
Useless spaces between words have been removed.
Reviewer 2 Report
The authors reviewed the direct interaction of several miRNAs and circRNAs with p53 protein. This review manuscript is well written; however, some revisions are required.
- Several images (miRNAs, complex between p53 and miRNAs, and circRNAs) were shown in this manuscript and these figures were clear and easy to understand. However, the regulation of p53 by these miRNAs and circRNAs was not clear. I recommend the authors to summarize the regulation (inhibition) of p53 by the miRNAs and circRNAs in a Table.
- “RNAse III enzyme”->”RNase III enzyme” (P3L36). “see Fig.7B”->”see Fig.8B?” (P14L2).
Author Response
- -Several images (miRNAs, complex between p53 and miRNAs, and circRNAs) were shown in this manuscript and these figures were clear and easy to understand. However, the regulation of p53 by these miRNAs and circRNAs was not clear. I recommend the authors to summarize the regulation (inhibition) of p53 by the miRNAs and circRNAs in a Table.
A Table which summarizes the proposed regulation as induced by the binding of miRNAs and circRNAs to p53 has been added to the manuscript (see new Table 1).
- “RNAse III enzyme”->”RNase III enzyme” (P3L36). “see Fig.7B”->”see Fig.8B?” (P14L2).
These typos have been corrected.
Reviewer 3 Report
The authors summarized the non-coding RNAs (miRNA and circRNA) that involved in the regulation networks of p53, the star molecular in cancer research. The paper is well-organised. However, I want to suggest one minor change about abbreviation.
Abbreviations must be spelled out at first mention in the paper. But the authors did not explain some of them at the first time, such as MDM2 and COP1. Furthermore, the authors did not give the explanation of some abbreviations in the paper, such like TUP and I-TASSER et al.
Author Response
Abbreviations must be spelled out at first mention in the paper. But the authors did not explain some of them at the first time, such as MDM2 and COP1. Furthermore, the authors did not give the explanation of some abbreviations in the paper, such like TUP and I-TASSER et al.
All the abbreviations have been now explained in the text.